SEPALLATA­-like genes of Isatis indigotica can affect the architecture of the inflorescences and the development of the floral organs

Ma Yan-Qin
Pu Zuo-Qian
Tan Xiao-Min
Meng Qi
Zhang Kai-Li
Yang Liu
Ma Ye-Ye
Huang Xuan xuanhuang@nwu.edu.cn
Xu Zi-Qin ziqinxu@nwu.edu.cn
College of Life Sciences, Northwest University , Xi’an, Shaanxi , China
Uversky Vladimir
Electronic publication date: 2022 Mar 1
Publication date: 2022
Volume: 10
Electronic Location ID: e13034
Received 2021 Sep 6; Accepted 2022 Feb 8
Copyright: © 2022 Ma et al.
Copyright year: 2022
Copyright holder: Ma et al.
License: This is an open access article distributed under the terms of the Creative Commons Attribution License, which permits unrestricted use, distribution, reproduction and adaptation in any medium and for any purpose provided that it is properly attributed. For attribution, the original author(s), title, publication source (PeerJ) and either DOI or URL of the article must be cited.
License URL: https://creativecommons.org/licenses/by/4.0/

Keywords: Isatis indigotica Fortune, Arabidopsis thaliana, SEPALLATA-like genes, Inflorescence architecture, Floral transition, Floral organ differentiation, Petal rescue, Floral meristem determinacy, Overexpression, Floral homeotic conversion

Funding: National Natural Science Foundation of China 31300223, 30870194, J1210063 Natural Sciences of Shaanxi Province 2021JZ-41 This work was supported by the National Natural Science Foundation of China (31300223, 30870194, J1210063), and the Major Project of Basic Research Program of Natural Sciences of Shaanxi Province (2021JZ-41). The funders had no role in study design, data collection and analysis, decision to publish, or preparation of the manuscript.

==============================
Background

The architecture of inflorescence and the development of floral organs can influence the yield of seeds and have a significant impact on plant propagation. E-class floral homeotic MADS-box genes exhibit important roles in regulation of floral transition and differentiation of floral organs. Woad (Isatis indigotica) possesses unique inflorescence, floral organs and fruit. However, very little research has been carried out to determine the function of MADS-box genes in this medicinal cruciferous plant species.

Results

SEPALLATA orthologs in I. indigotica were cloned by degenerate PCR. The sequence possessing the highest identity with SEP2 and SEP4 of Arabidopsis were named as IiSEP2 and IiSEP4, respectively. Constitutive expression of IiSEP2 in Columbia (Col-0) ecotype of Arabidopsis led to early flowering, and the number of the flowers and the lateral branches was reduced, indicating an alteration in architecture of the inflorescences. Moreover, the number of the floral organs was declined, the sepals were turned into carpelloid tissues bearing stigmatic papillae and ovules, and secondary flower could be produced in apetalous terminal flowers. In 35S::IiSEP4-GFP transgenic Arabidopsis plants in Landsberg erecta (Ler) genetic background, the number of the floral organs was decreased, sepals were converted into curly carpelloid structures, accompanied by generation of ovules. Simultaneously, the size of petals, stamens and siliques was diminished. In 35S::IiSEP4-GFP transgenic plants of apetalous ap1 cal double mutant in Ler genetic background, the cauliflower phenotype was attenuated significantly, and the petal formation could be rescued. Occasionally, chimeric organs composed of petaloid and sepaloid tissues, or petaloid and stamineous tissues, were produced in IiSEP4 transgenic plants of apl cal double mutant. It suggested that overexpression of IiSEP4 could restore the capacity in petal differentiation. Silencing of IiSEP4 by Virus-Induced Gene Silencing (VIGS) can delay the flowering time, and reduce the number and size of the floral organs in woad flowers.

Conclusion

All the results showed that SEPALLATA-like genes could influence the architecture of the inflorescence and the determinacy of the floral meristems, and was also related to development of the floral organs.

Introduction

The changing process of plant growth from vegetative stage to reproductive stage is known as floral transition and is associated with highly complicated regulatory networks. In the initial period of floral transition, the shoot apical meristem will expand in diameter and become inflorescence meristem gradually. In Arabidopsis, floral meristems will form continuously on the flank of inflorescence meristem and arrange spirally along the stem. In the course of subsequent growth, primordia of four floral organ types will appear in specific areas of floral meristems from outside to inside, and they will differentiate into sepals, petals, stamens and pistil, respectively (Smyth, Bowman & Meyerowitz, 1990).

MADS-box family genes play crucial roles in reproduction of flowering plants (Ma et al., 2020). In 1990’s, Coen and Meyerowitz proposed the “ABC model” for floral organ construction, which provided a theoretical basis for elucidation of Gothe’s intuitive idea that all floral organs have arisen from a modified leaf (Coen & Meyerowitz, 1991). The flowers in most angiosperms are composed of four whorls of floral organs: sepals, petals, stamens and carpels. The identity of these floral organs is determined by a specific combination of floral homeotic MADS-box transcriptional factors, as suggested in “floral quartet model” and ABCDE model (Pelaz et al., 2001; Honma & Goto, 2001; Theissen & Saedler, 2001; Shen et al., 2019). In the floral quartet model, complexes of MADS-box proteins are supposed to be involved in specifying the identity of different floral organs (Theissen & Saedler, 2001; Gramzow & Theissen, 2010).

In Arabidopsis, there are four E-class floral homeotic genes and they are named as SEPALLATA1 (SEP1, previously known as AGL2), SEP2 (AGL4), SEP3 (AGL9), and SEP4 (AGL3). Mutation in anyone of these SEP genes produces only subtle phenotypic changes, implying their functional redundancy (Pelaz et al., 2000). Consistent with this, the four different types of floral organs in sep1 sep2 sep3 triple mutant are converted into sepals, accompanied by loss of flower determinacy. More seriously, the flowers of sep1 sep2 sep3 sep4 quadruple mutant are comprised entirely of leaf-like structures (Pelaz et al., 2000).

Nonetheless, SEP1, SEP2 and SEP3 are not completely redundant with SEP4, owing to the different phenotypes between the triple mutant and the quadruple mutant. Further investigation revealed that SEP4 had a more prominent role in maintenance of the determinacy of floral meristems, because ap1 sep4 double mutants display a cauliflower-like phenotype which was observed neither in the respective single mutants nor in ap1 sep1 sep2 or ap1 sep3 mutants (Ditta et al., 2004). The single mutants of SEP3 produced sepaloid petals resemble to those of the intermediate ap1 mutants (Pelaz et al., 2001), indicating the redundancy of SEP1, SEP2 and SEP3 is not complete either. Besides, the dramatic floral defects of lfy sep3 double mutants are not found in either of the single mutants (Liu et al., 2009). It suggests that the phenotypic variations produced by loss-of-function allele of SEP3 could not be compensated by other SEP genes when LFY was mutagenized simultaneously.

The functional differences among SEP genes are probably associated with their inconsistency in expression. The transcripts of SEP1, SEP2 and SEP4 are detectable in floral meristems and in sepal primordia. In contrast, SEP3 is activated at the end of the second stage during flower development (Flanagan & Ma, 1994; Savidge, Rounsley & Yanofsky, 1995; Mandel & Yanofsky, 1998). Unlike other SEP genes, the transcription of SEP4 is not confined to flowers, the mRNA of this gene can also be detected in vegetative tissues, including leaves and stems (Ma, Yanofsky & Meyerowitz, 1991). It is very likely that these variations in expression are related to the diversified functions of the SEP genes (Jack, 2001).

A number of SEP-like genes have been characterized in other plant species. Down-regulation of GRCD1 (Gerbera Regulator of Capitulum Development 1), a homologous gene of SEP1 in Gerbera hybrida, converted the female florets into petals (Kotilainen et al., 2000). Inhibition of TM29, a tomato SEPALLATA homolog, resulted in aberrant phenotypes in the inner whorls of floral organs, resembling the cosuppression lines of FBP2 in petunia and TM5 in tomato, two other SEP-like genes (Ampomah-Dwamena et al., 2002). In cultivar ‘Ryokusei’ of Habenaria radiate, a mutant containing a retrotransposon-like element in the first exon of HrSEP-1, the white petals were changed into greenish petals, sepaloid tissues and a ventral column were formed in the inner whorls of flowers. Transcript of HrSEP-1 was undetectable in ‘Ryokusei’, namely loss-of-function of HrSEP-1 could influence the development of the floral organs (Mitoma & Kanno, 2018).

In five SEP-like genes of London plane (Platanus acerifolia), a basal eudicot tree, PlacSEP1.1, PlacSEP1.2 and PlacSEP1.3 belong to the SEP1/2/4 clade, PlacSEP3.1 and PlacSEP3.2 are members of SEP3 clade. The transcripts of PlacSEP1.3, PlacSEP3.1 and PlacSEP3.2 were detectable during the initiation stage of male and female inflorescences, and throughout the development of flowers and fruits, whereas PlacSEP1.2 was expressed only in female inflorescences. In addition, the expression of PlacSEP1.3 and PlacSEP3.1 was weak in vegetative buds of adult trees, implying their possible roles in regulation of dormancy. These results also showed that patterning of the inflorescences was associated with SEP-like genes. Except in cases of early flowering, overexpressing tobacco plants of PlacSEP1.1 and PlacSEP3.1 produced more lateral branches (Zhang et al., 2017a). Knockdown of SLMBP21, a tomato SEPALLATA-like gene in FBP9/23 subclade, could suppress the development of abscission zone in flower pedicels by affecting a subset of genes related to meristem activity, and overexpression of this gene could produce small cells at the proximal section of pedicels and peduncles (Liu et al., 2014).

Thalictrum thalictroides is a non-core eudicot species whose flowers are apetalous. By Virus-Induced Gene Silencing (VIGS), SEP-like genes in T. thalictroides were found to be partially redundant in specifying the identity of sepals and stamens, and in promoting petaloidy of sepals. Moreover, the ortholog of SEP3 showed a pronounced role in determination of carpel identity, and in carpel development. Furthermore, ThtSEP1 was involved in defining the boundary between sepals and stamens, and ThtSEP2 functioned in maintaining the boundary between stamens and pistil. In double-knockdown or triple-knockdown plants, partial to complete homeotic conversion of stamens and carpels to sepaloid organs or green sepals was occurred (Soza et al., 2016). Silencing of FaMADS9, a SEP1/2-like gene in strawberry, could inhibit the development of petals, and the ripening of achene and receptacle (Seymour et al., 2011).

In cucumber (Cucumis sativus L.), a mutant showing perturbations in the development of stamens, pistils and fruits was found to be related to CsSEP2. In this SEP-like gene, the change at the donor splice site in 5’-end of the sixth intron results in skipping of the sixth exon during RNA processing, which leads to loss of the transcriptional activation capacity of the protein encoded by the alternative transcript. The phenotypes of CsSEP2 mutant include enormously large sepals in female and male flowers, irregular shape or loss of three stigma lobes and three-fold longer style in female flowers, larger male flowers with separate filaments and an inner cavity filled by style-like tissues. Moreover, elongated carpel primordia can be observed before anthesis in female and male flowers of the mutant, accompanied by early shedding of the fruits (Wang et al., 2016). These results indicated that CsSEP2 was a homologous gene of Arabidopsis SEP2 and was involved in floral organ and fruit development in cucumber.

Woad (Isatis indigotica Fortune) is one of the traditional Chinese medicinal plants and is a biennial crucifer with bisexual flowers. Woad flower is composed of four whorls of floral organs: sepals, petals, stamens, and carpels. Up to now, a good deal of research has been devoted to examine the medicinal value of woad (Hu et al., 2011; Li et al., 2014), but the genes involved in regulation of flowering in this plant species are so far very rarely concerned. There are also four SEP-like genes in woad, and these genes were named as IiSEP1, IiSEP2, IiSEP3 and IiSEP4 according to the degree of similarity with the corresponding orthologous gene in Arabidopsis (Ma et al., 2019; Pu et al., 2020). In the present work, the function of IiSEP2 and IiSEP4 was investigated. To gain insight into the conservation of SEP activity among different species, transgenic Col-0 (Columbia) Arabidopsis plants overexpressing IiSEP2 were generated and analyzed. IiSEP4 isolated in previous work of our laboratory (Pu et al., 2020) was used in preparation of the ectopic expressing lines of wild-type Arabidopsis or ap1 cal double mutant in Ler (Landsberg erecta) genetic background. At the same time, silencing by VIGS was carried out to determine the function of IiSEP4.

Materials and Methods

Plant materials and bacterial strains

Woad plants, wild-type Arabidopsis ecotype Col-0, wild-type Arabidopsis ecotype Ler, and ap1 cal double mutant in Ler genetic background (CS67157) were used as materials in the present work, and were grown under normal greenhouse conditions (23 ± 2 °C, 16 h light/8 h dark). Woad flowers and leaves were used as materials in extraction of RNA and DNA. pMD 18-T was used in cloning of cDNA and genomic DNA fragments. pCAMBIA1302 was used in construction of the binary expression vectors. Escherichia coli strain DH5α and Agrobacterium tumefaciens strain GV3101 were used as host in isolation of IiSEP2 and in transformation of Arabidopsis, respectively.

Cloning of the coding sequence and the promoter of IiSEP2

Total RNA was extracted from floral organs of I. indigotica with Trizol reagent (Invitrogen, Waltham, MA, USA). Genomic DNA was eliminated by RNase-free DNase I (Takara, Tokyo, Japan). First strand cDNA was synthesized with PrimeScript 1st Strand cDNA Synthesis Kit (Takara, Tokyo, Japan). The coding sequence of IiSEP2 was obtained by PCR and the primers were designed according to the open reading frames of the MADS-box genes in different plant species (Table S1).

Promoter sequences were isolated with Universal Genome Walker 2.0 Kit (Clontech, Mountain View, CA, USA). The primers were designed according to the sequence in the first exon of IiSEP2 (Table S1). Genomic DNA was extracted from leaves of I. indigotica by CTAB method (Stewart & Via, 1993). Uncloned libraries were created after restriction digestion of the genomic DNA with Dra I, EcoR V, Pvu II or Stu I, and ligation of the adaptors provided by Universal Genome Walker 2.0 Kit (Clontech, Mountain View, CA, USA). Upstream sequences were amplified with Adaptor Primer 1 and IiSEP2-Upstream1 with Advantage 2 PCR Kit (Clontech, Mountain View, CA, USA). Nested PCR was carried out with Adaptor Primer 2 and IiSEP2-Upstream2. The PCR products were ligated into pMD 18-T and were sequenced after identification by colony PCR.

Analyses of the expression pattern of IiSEP 2 in I. indigotica

Total RNA was extracted from different tissues and floral organs of I. indigotica with RNAprep Pure Plant Kit (TIANGEN, Beijing, China) and the first strand cDNA was synthesized with PrimeScript 1st Strand cDNA Synthesis Kit (Takara, Tokyo, Japan). Quantitative real-time PCR (qRT-PCR) was conducted in BIO-RAD real-time PCR system (BIO-RAD, Hercules, CA, USA) with the primers shown in Table S1 (Huang et al., 2008; Ma et al., 2019). Woad actin (GenBank accession No. AY870652.1) was used as a reference gene. Triplicate qRT-PCRs were carried out and relative quantification of the transcript levels was accomplished using the comparative threshold cycle (Ct) method. Relative quantification refers to that the PCR signals of the IiSEP2 transcript in leaves, stem apexes, early inflorescences (the inflorescence shrinks into a ball), later inflorescences (the inflorescence is completely unfolded), flowers in full bloom, floral organs (include sepals, petals, stamens, and pistils), and young silicles are normalized to the PCR signal in roots. The fold change was calculated using the following formula: fold change = 2−∆∆Ct, where ∆∆Ct = (CtIiSEP2 − CtIiactin) different sample − (CtIiSEP2 − CtIiactin) root.

Subcellular localization by confocal laser scanning microscopy

The coding sequence of IiSEP2 was inserted into pCAMBIA1302 containing CaMV 35S promoter after high-fidelity amplification with the primers shown in Table S1. The recombinant construct was introduced into competent cells of A. tumefaciens GV3101 by freeze-thaw method (Höfgen & Willmitzer, 1988). Transformed A. tumefaciens cells were selected on YEB medium containing rifampicin (50 μg/L), gentamicin (50 μg/L), and kanamycin sulfate (50 μg/L). Positive colony carrying recombinant plasmid was propagated in liquid medium for 24 h at 30 °C in a shaker incubator (~200 rpm). Precipitated bacterial cells were resuspended with fresh YEB medium containing the same antibiotics and were grown to an OD600 value of 0.8. The bacterial cells were collected by centrifugation at 6,000 rpm for 10 min under room temperature. The pellet was resuspended with agroinfiltration buffer (10 mmol/L 2-(N-morpholino)ethanesulfonic acid, 200 μmol/L acetosyringone, 10 mmol/L MgCl2) to an OD600 value of 0.7. The bacterial suspension was infiltrated into the abaxial leaf epidermis of 6-week-old Nicotiana benthamiana plants using a syringe without needle. The quality of transient expression was estimated 60 h after agroinfiltration. Confocal laser scanning microscopy of the living plant tissues was performed using an OLYMPUS FLUOVIEW FV1000 microscope. Green fluorescence of GFP was observed with an exciting light of 488 nm.

Identification of IiSEP2 transgenic Arabidopsis plants

The recombinant binary expression vector pCAMBIA1302-IiSEP2, in which IiSEP2 is controlled by CaMV 35S promoter, was introduced into A. tumefaciens GV3101. Agrobacterium-mediated transformation of Arabidopsis was performed essentially according to the protocol of Clough & Bent (1998). Col-0 transgenic seedlings were selected on hygromycin-containing medium, and the expression levels of IiSEP2 in T2 generation were detected by RT-PCR using IiSEP2 specific primers (Table S1), and by Western blotting using rabbit monoclonal antibody of GFP (Sino Biological Inc., Beijing, China). Transgenic phenotypes were confirmed in T1 and T2 generations. Influence of IiSEP2 on Arabidopsis MADS-box genes, including AP3 (APETALA3), PI (PISTILLATA), SHP1 (SHATTERPROOF1) and SHP2, was determined by qRT-PCR.

Preparation of IiSEP4 transgenic plants in the genetic background of Ler

The binary expression vector pCAMBIA1302-IiSEP4 constructed in our previous work (Pu et al., 2020) was introduced into A. tumefaciens GV3101 and was used in genetic transformation of wild-type Arabidopsis plants (Ler) or ap1 cal double mutant (Ler). The expression of the transgene was analyzed by qRT-PCR with IiSEP4-qRT-F (5′-AGATAGCCGGGATGGGAGTG-3′) and IiSEP4-qRT-R (5′-AATCATCGACCGGGCCTTTG-3′) (Pu et al., 2020). To compare the phenotypic differences, 25 IiSEP4 transgenic Arabidopsis plants were grown in greenhouse together with 25 wild-type plants every time.

Silencing of IiSEP4 by VIGS

To investigate the function of IiSEP4 in woad, its expression was downregulated by Agrobacterium-mediated and TRV-based (tobacco rattle virus-based) VIGS (Liu, Schiff & Dinesh-Kumar, 2002). A. tumefaciens GV3101 was cultured in LB liquid medium containing 50 mg/L kanamycin, 25 mg/L rifampicin and 25 mg/L gentamicin to OD600 = 2.0. After centrifugation, the bacterial cells were resuspended with infiltration solution (100 mL was composed of 100 µL 0.2 mol/L acetosyringone, 2 mL 0.5 mol/L 2-(N-morpholino)ethanesulfonic acid, 1 mL 1 mol/L MgCl2). During four-leaf stage of growth, the leaves of woad plants were infiltrated with a mix of A. tumefaciens GV3101 carrying pTRV1 and A. tumefaciens GV3101 carrying recombinant pTRV2-IiSEP4 containing a 300 bp specific fragment coming from 5’-end of IiSEP4 coding sequence (Pu et al., 2020). The fragment of IiSEP4 was amplified by high-fidelity PCR with primers VIGS-IiSEP4-F (5′-CGCGAATTCCGATGATTGATCAACTATCG-3′) and VIGS-IiSEP4-R (5′-CGCGGATCCTCTGGGATGTTGTTGCAGAG-3′). In each VIGS experiment, the phenotypes of 30 woad plants in treatment group and 30 woad plants in control group were compared.

Data analysis

SPSS 22.0 (SPSS Inc., Chicago, IL, USA) was used for data analysis. The data of rosette leaf number, the angle between cauline leaf and stem, flower bud number, stamen number in IiSEP2 overexpressing Arabidopsis plants, and the data about the expression pattern of IiSPE2 in woad plants, were analyzed by one-way ANOVA, followed by Tukey’s post-hoc test. The data obtained in qRT-PCR analysis of the MADS-box genes (including AP3, PI, SHP1 and SHP2) in IiSEP2 overexpressing Arabidopsis plants, IiSEP4 in 35S::IiSEP4-GFP transgenic plants in wild-type Ler genetic background and in 35S::IiSEP4-GFP transgenic plants of ap1 cal double mutant in Ler genetic background, and IiSEP4 in distal noninfiltrated leaves in woad plants infiltrated with pTRV1 + pTRV2-IiSEP4, were analyzed by two-sided Student’s t-test (Pu et al., 2020).

Results

Identification of the coding sequence and the promoter of IiSEP2

The coding sequence of IiSEP2 was amplified with high-fidelity DNA polymerase by RT-PCR. Sequencing result showed that the open reading frame of IiSEP2 was 753 bp in length and encoded a 250-amino-acid protein (Fig. S1). IiSEP2 comprises the four regions of typical plant MIKC-type MADS-box proteins, namely MADS domain (M, 2-60aa), intervening region (I, 61-83aa), keratin-like domain (K, 84-171aa), and C-terminal region (C, 172-250aa). At the same time, the protein encoded by IiSEP2 shows high identity in amino acid sequence with the homologous proteins of other plant species. For instance, the identity between IiSEP2 and Arabidopsis SEP2 is 95% (Fig. S2A). Eight conserved motifs with a length of 6~50 amino acids could be found in IiSEP1, IiSEP2 and IiSEP2-like proteins by the online tool in MEME website (http://meme-suite.org/) (Bailey et al., 2009), and two motifs (Motif 1 and Motif 9) in MADS domain and three motifs (Motif 3, Motif 4 and Motif 5) in K-box domain existed in all the analyzed proteins (Fig. S2B). The logo of these motifs discovered by MEME is shown in Fig. S2C.

To clarify the relationship of IiSEP2 and other plant MADS-box proteins, we analyzed the phyletic evolution of IiSEP2 (Fig. S3). The results showed that IiSEP2 was closely related to Arabidopsis SEP2. These results demonstrated that IiSEP2 was an E-class floral homeotic gene and further confirmed that IiSEP2 was the orthologous gene of Arabidopsis SEP2.

In 1 kb regulatory sequence upstream of the initiation codon characterized by Genome Walking method (Sequence Data S1), there are three CArG-boxes, indicating the expression of IiSEP2 can be affected by other MADS-box transcriptional factors.

Spatial and temporal expression of IiSEP 2 in I. indigotica

The abundances of IiSEP2 mRNA in various tissues and floral organs of I. indigotica were analyzed by qRT-PCR. The transcriptional levels of IiSEP2 were low in roots and leaves. However, the quantity of IiSEP2 transcript was significantly increased in stem apex and was accumulated to a high level in flowers and young silicles. Further testing in floral organs showed that IiSEP2 mRNA was highly expressed in sepals, petals, stamens, and pistil (Fig. 1). Taken together, the qRT-PCR results illustrate that IiSEP2 expression occurs from the early stages of flower development to young silicles.

Figure 1 Expression pattern of IiSEP2 in I. indigotica analyzed by qRT-PCR.

Error bars represent the standard deviation. Root was used as normalization sample. Multiple testing correction was conducted with SPSS software, different letters indicate significant difference at 5% level.

Subcellular localization of IiSEP2

To determine the subcellular localization of IiSEP2 in vivo, the coding sequence of IiSEP2 without the stop codon was fused in frame with GFP reporter gene in pCAMBIA1302 and was driven by CaMV 35S promoter. A. tumefaciens carrying the recombinant expression vector was used to perform a transient expression assay in N. benthamiana leaves. The GFP fluorescence of the control experiment using empty vector evenly distributed throughout the cell, whereas the GFP fluorescence of IiSEP2-GFP was observed only in the nucleus (Fig. S4). The result is consistent with the predictions using PSORT II online server (http://psort.hgc.jp/form2.html), namely KRIENKINRQVTFAKRR in N-terminus of IiSEP2 is a bipartite nuclear localization signal (NLS).

Overexpression of IiSEP2 leads to morphologic changes in transgenic Arabidopsis plants

To examine the function of IiSEP2, transgenic Arabidopsis plants under the control of the constitutive CaMV 35S promoter were prepared in this work. Seeds were selected on MS medium containing hygromycin, and the putative transgenic plants were confirmed by PCR and Western blotting (Fig. S5).

In 28 35S::IiSEP2 independent transgenic lines, nine transgenic lines showed obvious macroscopic differences in comparison with the wild-type Arabidopsis plants. 35S::IiSEP2 transgenic Arabidopsis plants were extremely early in bolting. The flowering time was evaluated by counting the number of the leaves produced when the first flower bloomed. The transgenic lines generated 4–5 leaves prior to the onset of flowering, whereas wild-type plants produced approximately nine leaves (Fig. 2A).

Figure 2 Effects of IiSEP2 overexpression on aerial architecture of Arabidopsis plants.

(A) Comparison of rosette leaf number produced by 35S::IiSEP2 transgenic and wild-type Col-0 plants before flowering. (B) Comparison of the angle between the cauline leaves and the stems. (C) Comparison of the flower bud number produced by 35S::IiSEP2 transgenic and wild-type Col-0 plants. Values correspond to mean ± standard error (n = 15). Multiple testing correction was conducted with SPSS software; different letters indicate significant difference at 5% level.

In wild-type Col-0 Arabidopsis plants, cauline leaves are generated at the lower part of the inflorescence stems, and the oval-shaped blades are slightly curved outward. Moreover, the angles between the cauline leaves and the stems in wild-type Col-0 Arabidopsis plants are larger (Figs. 2B, 3A and 3B). Different from wild-type Col-0 Arabidopsis plants, the cauline leaves of IiSEP2 transgenic Arabidopsis plants are clearly curling inward and the margin is involute. Simultaneously, the angles between the cauline leaves and the stems become smaller in transgenic Arabidopsis plants, and the whole blade looks like an inverted cone (Figs. 2B and 3C–3G).

Figure 3 Morphological comparison of the cauline leaves between wild-type Col-0 and IiSEP2 transgenic plants.

(A) The rosette leaves and cauline leaves of wild-type Col-0 plants, the angles between the oval-shaped cauline leaves and the stems are larger. (B) The upper part of a wild-type Col-0 plant in the flowering stage. (C) IiSEP2 transgenic Arabidopsis plants. (D) IiSEP2-OE-6#. (E) IiSEP2-OE-7#. (F) IiSEP2-OE-8#. (G) IiSEP2-OE-9#. IiSEP2 transgenic Arabidopsis plants in (F) and (G) show the inward-curling cauline leaves, and the angles between the cauline leaves and the stems are reduced. In (A–C), scale bars represent 1 cm. In (D–G), scale bars represent 1 mm.

The process that the plant switches from vegetative growth to reproductive growth is named as floral transition. At the early stage of floral transition, the vegetative shoot apical meristem is transformed into an inflorescence meristem, and floral meristems are formed in the peripheral regions of the inflorescence meristem. Then the primordia of the floral organs will be produced along with the development of floral meristems, which is accompanied by definition of the boundaries between different floral organs. The inflorescence meristem of wild-type Col-0 Arabidopsis plants is highly active during the growth process and can produce a lot of flower buds continuously (Figs. 4A and 4B). However, the activity of the inflorescence meristem in IiSEP2 transgenic Arabidopsis plants is reduced remarkably. In general, only a few flower buds could be generated at the top of the inflorescence stem in IiSEP2 transgenic Arabidopsis plants (Figs. 2C and 4C–4F), illustrating that constitutive expression of IiSEP2 could reduce the number of flowers and the number of lateral branches. Namely, IiSEP2 can affect the architecture of the inflorescences.

Figure 4 Influence of IiSEP2 on development of inflorescences.

(A & B) Indefinite inflorescences of wild-type Col-0 plants. The inflorescence meristem could produce new flower buds continuously. (C) Inflorescence of IiSEP2 overexpressing Arabidopsis plant, only a few flower buds could be observed. (D) Wilted inflorescence with undersized flower buds at early development stage in IiSEP2 transgenic Arabidopsis plant. (E & F) The inflorescences converted into defective terminal flowers in IiSEP2 transgenic Arabidopsis plant, arrow indicate stamens. (G) Comparison of the number of stamens per flower. Scale bars represent 1 mm. Multiple testing correction was conducted with SPSS software, different letters indicate significant difference at 5% level.

Under normal circumstances, the flowers of the wild-type Col-0 Arabidopsis plants consist of four whorls of floral organs, including four sepals in the outermost whorl, four petals in the second whorl, six stamens in the third whorl, and a gynoecium in the fourth whorl (Figs. 5A–5C). By observation under stereomicroscope, it was found that overexpression of IiSEP2 in Arabidopsis resulted in seriously anomalous development of the floral organs. A number of flowers were male sterile (Fig. 4D), owing to the abnormality of stamens in development. At the same time, the inflorescence meristems were transformed into defective terminal flowers without sepals and petals, and the number of stamens was reduced (Figs. 4E–4G). In Fig. 4E, only three stamens could be observed under the pistil in the middle. In many flowers, the sepals were converted homeotically into carpelloid structures with stigmatic papillae and ovules in transgenic lines 7 and 8 (Figs. 5E and 5J).

Figure 5 Abnormal phenotypes of various floral organs induced by IiSEP2 overexpression.

(A–C) The flowers of wild-type Col-0 Arabidopsis plants. The sepals and petals have been removed from Arabidopsis flower in (C). (D) IiSEP2 transgenic line 3, red arrow indicate terminal flower containing a secondary flower, blue arrow indicate lateral flower. (E) IiSEP2 transgenic line 7, the sepals were converted into carpelloid tissues. (F) IiSEP2 transgenic line 4, red arrow indicate terminal flower containing a secondary flower. (G) IiSEP2 transgenic line 5, five petals could be observed. (H) IiSEP2 transgenic line 6, apetalous flower was generated. (I) Terminal flowers and secondary flower in crossing progeny of IiSEP2 transgenic line 8 and wild-type Col-0. (J) IiSEP2 transgenic line 8, abnormal flowers were produced. (K) Flower lacking petal in crossing progeny of IiSEP2 transgenic line 8 and wild-type Col-0 plant. (L) Abnormal floral organs in crossing progeny of IiSEP2 transgenic line 8 and wild-type Col-0 plant. Scale bars represent 1 mm.

In transgenic lines 3 and 4, secondary flower could be generated in terminal flowers. The number of sepals, petals and stamens was decreased in these terminal flowers, whereas the secondary flowers were nearly normal in structure (Figs. 5D and 5F). Different from four petals in wild-type flowers, a few apical flowers of the IiSEP2 overexpressing plants produced five petals or lacked petal, together with anomalous development of the carpels in transgenic lines 5 and 6 (Figs. 5G and 5H). Quantitative expression analyses showed that AP3 and PI, two Arabidopsis MADS-box genes related to petal formation, were suppressed by ectopic expression of IiSEP2 (Fig. S6). On the contrary, the transcripts of SHP1 and SHP2 were increased in IiSEP2 transgenic Arabidopsis plants.

In order to fully understand the function of IiSEP2, transgenic line 8 was genetically crossed with the wild-type Col-0 plants, and the phenotypes of the crossing progenies were compared to that of the IiSEP2 transgenic line 8. The phenotypes of the IiSEP2 transgenic lines were maintained in filial generations of the crossing lines. These offsprings exhibited the phenotypic variations of the IiSEP2 overexpressing plants, including the secondary flowers, the loss of petals, and the reduction of stamens (Figs. 5I, 5K, and 5L). All these phenotypic changes indicated that constitutive expression of IiSEP2 could affect the morphogenesis of the flowers.

Ectopic expression of IiSEP4 in Ler genetic background can affect the development of floral organs

In order to analyze the regulatory function of IiSEP4 in floral transition and flower development, transgenic Arabidopsis plants were prepared. The transcriptional levels of IiSEP4 in transgenic plants were detected by qRT-PCR. The results showed that IiSEP4 could express effectively in transgenic plants of wild-type Arabidopsis in Ler genetic background, and in transgenic plants of ap1 cal double mutant in Ler genetic background (Fig. S7).

All the 35S::IiSEP4-GFP transgenic plants displayed abnormal phenotypes, and the number of the floral organs was reduced. In particular, sepals were curly and were converted into carpelloid structures, and were obviously accompanied by the generation of ovules (Figs. 6B and 6C). In addition, dwarf and abnormal petals and stamens were produced (Fig. 6D), and the size of the silique was also decreased significantly in comparison with the wild-type Ler plants (Fig. 6F).

Figure 6 Phenotypic variations of the floral organs in IiSEP4 transgenic Arabidopsis (in Ler genetic background).

(A) The flower of the wild-type Ler plant. (B–E) The abnormal phenotype of the flowers in 35S::IiSEP4-GFP transgenic plants. (F) The silique in 35S::IiSEP4-GFP transgenic plant (left) and wild-type Ler plant (right). Bar = 1 mm.

On account of the apetalous phenotype of ap1 cal double mutant, ectopic expression in this mutational material was conducted to clarify the roles of IiSEP4 in petal differentiation. In ap1 cal double mutant in Ler genetic background, floral meristems were transformed into inflorescence meristems, and new meristems were elaborated continuously. As a result, the inflorescence resembles a cauliflower in ap1 cal double mutant. In addition, ap1 cal plants produced apetalous flowers, namely only sepals, stamens and pistil could be formed (Figs. 7A and 7B). In 35S::IiSEP4-GFP transgenic plants of ap1 cal double mutant, the cauliflower phenotype was attenuated significantly, and the petals could be recovered (Figs. 7C–7H). Occasionally, chimeric organ composed of petaloid and sepaloid tissues (Figs. 7F and 7G), and chimeric organ composed of petaloid and stamineous tissues (Fig. 7E), could be observed.

Figure 7 Phenotypic variations of IiSEP4 transgenic plants of ap1 cal double mutant in Ler genetic background.

(A & B) The inflorescence of ap1 cal double mutant. (C–H) The inflorescence of IiSEP4 transgenic plants of ap1 cal double mutant. (C) shows an incomplete cauliflower phenotype and petals reappeared. (D) is an enlarged photo of (C), a number of sepals, stamens and pistils can be found in one flower; In comparision with ap1 cal double mutant, multiple petals can be produced. (E) is the photo of the flower in (D) taken in different direction, arrow shows a chimeric organ composed of petaloid and stamineous tissues, representing fusion of petal and stamen. In (F and G), arrows show the chimeric organ composed of petaloid and sepaloid tissues, representing fusion of petal and sepal. (G and H) are the photos of the flower in (F) taken in different directions. Bar = 1 mm.

Phenotypic variations of woad plants after downregulation of IiSEP4 by VIGS

To analyze the moderating effects on woad flowering and to verify the results obtained in Arabidopsis, the expression level of IiSEP4 was downregulated by VIGS mediated by Agrobacterium. The results of RT-PCR showed that RNA molecules of TRV1 and TRV2 could be detected in distal noninfiltrated leaves of the control woad plants treated with pTRV1 + pTRV2, and in distal noninfiltrated leaves of the woad plants treated with pTRV1 + pTRV2-IiSEP4 (Fig. S8). qRT-PCR data showed that the mRNA abundance of IiSEP4 was reduced dramatically in woad plants treated with pTRV1 + pTRV2-IiSEP4 (Fig. S9), indicating IiSEP4 could be silenced effectively by VIGS.

Silencing of IiSEP4 can delay the flowering time. Compared to woad plants infiltrated with a mixture of A. tumefaciens GV3101 carrying pTRV1 and A. tumefaciens GV3101 carrying the empty pTRV2, the woad plants infiltrated with a mixture of A. tumefaciens GV3101 carrying pTRV1 and A. tumefaciens GV3101 carrying pTRV2-IiSEP4 start bolting nearly half a month later, indicating downregulation of IiSEP4 can lead to late-flowering phenotype in woad. Specifically, wild-type woad plants begin to bloom in first ten-days of May, whereas IiSEP4-silenced lines bear flowers in late May.

More than 90% of the woad plants in treatment group showed phenotypic variations. Observation of the flowers showed that the floral organs were unchanged after infiltration with pTRV1 + pTRV2 (Fig. 8A), whereas the floral organs in woad plants infiltrated with pTRV1 + pTRV2-IiSEP4 presented pronounced anomalous phenotypes, and the number of sepals, petals and stamens was reduced (Figs. 8B–8E). In comparison with the control group, the size of the floral organs in woad plants infiltrated with pTRV1 + pTRV2-IiSEP4 was much smaller (Figs. 8F–8H). In some woad plants treated with pTRV1 + pTRV2-IiSEP4, the number of petals was increased. For instance, one flower in a woad plant infiltrated with pTRV1 + pTRV2-IiSEP4 produced five petals, but the number of stamens was reduced from six to five. In these plants, except the smaller sepals, no obvious change could be observed in the size of petals and stamens (Figs. 8I–8L).

Figure 8 Phenotypic observation of the floral organs in woad plants treated with TRV-IiSEP4.

(A) Wild-type woad flower. (B–L) Phenotypic variations of the flowers in woad plants infiltrated with TRV-IiSEP4. (B) a flower constituted by one sepal, one petal, one stamen and one pistil. (C and D) anatomy view of the flower in (B). (E) a flower with three sepals, three petals, three stamens and one pistil. (F–H) comparison of the size of the floral organs in the flower of (E) (up) with the floral organs in wild-type flower (down). (I) a flower contains four sepals, five petals, five stamens and one pistil. (J–L) comparison of the size of the floral organs in the flower of (I) (up) with the floral organs in wild-type flower (down). Bar = 1 mm.

Discussion

Although A-, B- and C-class floral homeotic MADS-box genes are essential to the development of flowers, ectopic expression of these genes is not enough to completely induce the formation of floral organs, indicating other transcriptional factors are involved in floral patterning regulation (Krizek & Meyerowitz, 1996). Phenotypic observation of the multiple mutant showed that SEPALLATA1/2/3/4 (SEP1/2/3/4) are also necessary for differentiation of the four types of floral organs, and for initiation of floral meristems (Pelaz et al., 2000; Ditta et al., 2004). Simultaneous mutation of SEP1/2/3 converts the four types of floral organs into sepaloid tissues (Pelaz et al., 2000). If SEP1/2/3/4 are all knocked out, the four types of floral organs will be reversed as leaf-like tissues bearing epidermal trichomes, representing the functions of the E-class floral homeotic genes are completely destroyed (Ditta et al., 2004).

Flower development of Arabidopsis from initiation until the opening was divided into 12 stages (Smyth, Bowman & Meyerowitz, 1990). SEP2 was expressed in the entire floral meristems at stage 2, slightly earlier than SEP3. In subsequent stages, SEP2 was expressed in all the four types of floral organs (Pelaz et al., 2000; Ditta et al., 2004). ThtSEP2, a SEP2 homologous gene characterized in T. thalictroides, was expressed at low levels in all the floral organs and at all developmental stages of flowers (Soza et al., 2016). In the present work, the coding sequence of IiSEP2 was cloned by degenerate PCR and its length is 753 bp, and the MADS-box protein encoded by IiSEP2 consists of 250 amino acid residues. Multiple alignments showed that MADS-box proteins were highly conserved and the identity between IiSEP2 and Arabidopsis SEP2 was 95%. According to the results of qRT-PCR, IiSEP2 was expressed in early stage of flower development, all the four types of floral organs, and silicles. In four types of floral organs, IiSEP2 was highly expressed in sepals and petals. On the contrary, IiSEP2 mRNA could not be detected obviously in vegetative tissues. These results illuminate that IiSEP2 plays an ubiquitous and crucial role in flower development of I. indigotica.

In Arabidopsis, SEP1 and SEP2 are target genes of FAR-RED ELONGATED HYPOCOTYL3 (FHY3). FHY3 is associated with maintenance of shoot apical meristem, determinacy of floral meristem, and differentiation of petals and stamens. SEP2 can be activated directly by FHY3, and ultimately promote the determinacy of floral meristem (Li et al., 2016). In fhy3 single mutant, the pistil from the indeterminate floral meristem could generate plenty of floral organs. In inflorescences of fhy3-68 ag-10 double mutant, the transcripts of SEP1 and SEP2 were decreased. On the contrary, SEP3 expression was upregulated in fhy3-68 single mutant and unchanged in fhy3-68 ag-10 double mutant (Li et al., 2016). In consistent with this, ectopic expression of IiSEP2 in Arabidopsis can also influence the normal development of the floral organs and the siliques. Similar to 35S::IiSEP4-GFP transgenic plants in wild-type Ler genetic background obtained in the present work (Fig. 6F), FHY3 mutants in the genetic background of ag-10 produced very short and bulged siliques with an increased carpel number, accompanied by formation of very small petals and sterile anthers. IiSEP2 and IiSEP4 of I. indigotica probably also play an important role in determinacy of the floral meristems, and are controlled by the ortholog of FHY3 in I. indigotica.

Ectopic expression of SEP2 could rescue the defects of fhy3-68 ag-10 double mutant in floral meristem determinacy. In 35S:SEP2 fhy3-68 ag-10 transgenic Arabidopsis plants, normal siliques composed of gynophore and two fused carpels could be produced. It suggested that the function of FHY3 in floral meristem determinacy was mediated by SEP2. However, the other phenotypic variations in fhy3-68 ag-10 double mutant, including small inflorescence, short petal and sterile anther, could not be rescued by SEP2. Different from SEP2, overexpression of SEP1 in fhy3-68 ag-10 double mutant under the control of CaMV 35S promoter could not recover the defective phenotypes of floral meristems, indicating SEP2 possesses an unique role in pistil development and in determinacy of flowers. When SEP2 was driven by the flower-specific SEP3 promoter, its expression in fhy3-68 ag-10 double mutant could also rescue the floral meristem indeterminacy. Nonetheless, short siliques with multiple carpels and additional organs, and sterile anthers could be produced. When SEP2 was down-regulated by artificial miRNA in the genetic background of ag-10, bulged siliques with additional organs growing inside were generated, similar to the phenotype of fhy3-68 ag-10 double mutant, indicating suppression of SEP2 can enhance the indeterminacy of floral meristem (Li et al., 2016). IiSEP2 and IiSEP4 may also possess divergent or specified functions in comparison with other SEP-like genes in I. indigotica.

SEP-like genes were found to be involved in regulation of the inflorescence architecture. In sep1sep2sep3 triple mutant and sep1sep2sep3sep4 quadruple mutant, the degree of inflorescence branching was increased (Pelaz et al., 2000; Ditta et al., 2004). The transcript of SlCMB1, a SEP-like gene of tomato, was mainly accumulated in sepals. In RNA interference lines of SlCMB1, longer peduncles and more branches were generated. Moreover, generation of new leaves and apical meristem in RNA interference lines represents a loss of the inflorescence determinacy and a transition from reproductive growth to vegetative growth. In particular, abnormally fused and enlarged sepals could be observed in RNA interference lines (Zhang et al., 2018). In Gerbera hybrida, SEPALLATA-like MADS-box genes were associated with patterning regulation of the pseudanthial inflorescence (Zhang et al., 2017b). In monocotyledons, SEPALLATA-like genes could also influence the architecture of inflorescences (Gao et al., 2010; Lin et al., 2014; Song et al., 2018). In the present work, it was confirmed that overexpression of IiSEP2 and IiSEP4 in Arabidopsis resulted in reduction of the flower number and the lateral branch number.

TM5 is a SEP-like gene in tomato and is expressed at high levels in meristematic territories fated to form petals, stamens, and gynoecia. Suppression of TM5 resulted in morphogenetic alterations in the internal three whorls of floral organs. In antisense transgenic plants of TM5, the petals were green throughout the life span of the flowers, the anthers were also green and were converted into sepaloid or petaloid structures. Moreover, incomplete fusion of carpels and failure to form normal style could be observed (Pnueli et al., 1994). As a SEP-like gene in rice, severe loss-of-function mutations in OsMADS1 could cause complete homeotic conversion of lodicules, stamens and carpels into lemma-like or palea-like structures (Agrawal et al., 2005). PlacSEP1.1, PlacSEP1.2 and PlacSEP1.3 of London plane have been characterized functionally. Overexpression of PlacSEP1.1, PlacSEP1.2 and PlacSEP1.3 in Arabidopsis resulted in different phenotypic alterations. 35S:PlacSEP1.1 transgenic plants with the small size and curled leaves were early flowering obviously. 35S:PlacSEP1.3 transgenic lines also exhibited early flowering phenotype, and possessed less rosette leaves and more cauline leaves in comparison with the wild-type Arabidopsis plants. However, curled leaf could not be found. On the contrary, 35S:PlacSEP1.2 transgenic plants showed no visible phenotypic changes (Zhang et al., 2017a). TaMADS1 is a SEP-like gene isolated from wheat (Triticum aestivum L.). Compared to wild-type Arabidopsis, 35S:TaMADS1 transgenic lines were early flowering, the number of petals and stamens was reduced, some flowers had short filament and sterile petaloid anthers (Zhao, Cheng & Zhang, 2006). All these reports together with the results in the present work indicated that SEP-like genes were conserved in regulation of the development of floral organs in both dicotyledons and monocotyledons.

In this work, influence of IiSEP2 overexpression on MADS-box genes of Arabidopsis was also analyzed, including AP3, PI, SHP1 and SHP2. AP3 and PI are B-class floral homeotic MADS-box genes and are responsible for specifying the identity of petals. SHP1 and SHP2 are associated with formation of the dehiscence zone in siliques and development of ovules (Pinyopich et al., 2003). The results showed that AP3 and PI were downregulated in IiSEP2 transgenic Arabidopsis plants, whereas SHP1 and SHP2 were upregulated. Similar to IiSEP2 and IiSEP4, constitutive expression of SHP in Arabidopsis also resulted in ectopic formation of carpels and ovules (Favaro et al., 2003). In like manner, when LMADS2, a carpel-specific MADS-box gene of Lilium longiflorum, was constitutively expressed in Arabidopsis, sepals would be converted homeotically into carpelloid structures with stigmatic papillae and ovules (Tzeng, Chen & Yang, 2002).

Moreover, the apical flowers in IiSEP2 transgenic Arabidopsis plants often produced five petals, three pistils, shrunken stamens and secondary flowers, or apetalous. These data showed that the development of petals and stamens in Arabidopsis could be affected by ectopic expression of SEP-like genes, and phenotypic changes similar to ap3-3 and pi-1 mutants could be produced (Bowman et al., 1999). Quantitative real-time PCR showed that the expression of AP3 and PI could be inhibited by IiSEP2. However, ectopic expression of IiSEP4 could restore the development of petals in ap1 cal double mutant.

Except for the influences on flower development, IiSEP2 and IiSEP4 can affect the growth of leaves. The cauline leaves in IiSEP2 transgenic Arabidopsis are shaped like inverted cones, and the angles between the cauline leaves and the stems were reduced, which was probably associated with CURLY LEAF (CLF). The protein encoded by CLF was a subunit of polycomb repressive complex 2 (PRC2). CLF could change the transcriptional activity of the target genes by trimethylation of lysine 27 on histone H3 (H3K27me3) with its methyltransferase activity, and resulted in repression of the target genes (Ré et al., 2020; Singkaravanit-Ogawa et al., 2021). clf mutants showed leaf curling and early flowering phenotypes, which could be suppressed by mutation of SEP3. It has been reported that derepression of SEP3 could cause alterations in morphology of leaves (Mateo-Bonmatí et al., 2018). In clf mutant, SEP3 was highly expressed in leaves. Consistent with this, ectopic expression of SEP3 driven by CaMV 35S promoter produced curled rosette leaves (Pelaz et al., 2001), and overexpression of SEP-like genes can also alter the morphology of leaves (Adal et al., 2021).

Conclusions

In Arabidopsis, ectopic expression of IiSEP2 could lead to early flowering. In woad, downregulation of IiSEP4 by VIGS could produce late-flowering phenotypes. In comparison with the wild-type plants, transgenic Arabidopsis plants of IiSEP2 and IiSEP4 were relatively small and the main stems of these plants were thin. Constitutive expression of IiSEP2 and IiSEP4 in Arabidopsis could disturb the indeterminacy of inflorescence meristems, and only a small number of flower buds would be produced, accompanied by generation of terminal flower. Furthermore, IiSEP2 and IiSEP4 can influence the development of floral organs. In transgenic plants overexpressing IiSEP2 and IiSEP4, abnormal floral structures without sepals and petals were formed, and the number of stamens was reduced. Meanwhile, carpelloid structures with stigmatic papillae and ovules could be observed. In IiSEP4 transgenic plants in the genetic background of ap1 cal double mutant, petal formation could be rescued.

Supplemental Information

Supplemental Information 1 Supplementary Table, Figures and Sequence data.

Click here for additional data file.

Supplemental Information 2 Raw Data and original gel images-revision.

Click here for additional data file.

Additional Information and Declarations

Competing Interests

Author Contributions

Data Availability

The authors declare that they have no competing interests.

Yan-Qin Ma conceived and designed the experiments, performed the experiments, analyzed the data, prepared figures and/or tables, authored or reviewed drafts of the paper, and approved the final draft.

Zuo-Qian Pu conceived and designed the experiments, performed the experiments, analyzed the data, prepared figures and/or tables, authored or reviewed drafts of the paper, and approved the final draft.

Xiao-Min Tan performed the experiments, prepared figures and/or tables, and approved the final draft.

Qi Meng performed the experiments, prepared figures and/or tables, and approved the final draft.

Kai-Li Zhang performed the experiments, prepared figures and/or tables, and approved the final draft.

Liu Yang performed the experiments, prepared figures and/or tables, and approved the final draft.

Ye-Ye Ma performed the experiments, prepared figures and/or tables, and approved the final draft.

Xuan Huang analyzed the data, prepared figures and/or tables, and approved the final draft.

Zi-Qin Xu conceived and designed the experiments, authored or reviewed drafts of the paper, and approved the final draft.

The following information was supplied regarding data availability:

The raw data and full-length gel images are available in the Supplemental Files.

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
