# Peer review of "SEPALLATA­-like genes of Isatis indigotica can affect the architecture of the inflorescences and the development of the floral organs"

_PeerJ, doi:10.7717/peerj.13034_

## Round 0.1 · original submission · Major Revisions

As you can see, some serious concerns were raised by the reviewers. Please address critiques and amen manuscript accordingly.

Reviewer 1 ·

Basic reporting

I have no significant objections to the present article using these 4 objective criteria.

Experimental design

In this work, the authors sequence the likely orthologs of two SEPALLATA genes from the crucifer species Isatis indigotica (woad). They perform qRT-PCR studies to quantify the expression of one of the two cloned genes (IiSEP2) in floral and other organs of woad, and they also perform over-expression studies of this gene in transgenic Arabidopsis thaliana. The DNA sequences and RT-PCR results presented in this work are valid scientific data which add incrementally to the huge mass of data currently available on MADS-box genes in flowering plants. The studies performed in the present work are, however, extremely limited, notably as there are four SEP genes in Arabidopsis- so are there also SEP1 and SEP3 orthologs in woad, or have these genes been lost in the woad lineage? This might be an interesting question, but the present authors fail to address it. Also, the qRT-PCR results presented here reveal only that the woad ortholog of SEP2 is expressed in all floral organs, as is its Arabidopsis ortholog. This is hardly surprising, and provides no insight into the phenotypes (petal color, seed number, fruit morphology etc) cited as important differences between Arabidopsis and woad. In particular, the overexpression experiments perfomed here do not provide any information to significantly elucidate the role of SEP2 in woad. To clarify what I mean by this: if Hypothesis 1 was that SEP2 in woad performed exactly the same functions as SEP2 in Arabidopsis, and if Hypothesis 2 was that SEP2 in woad performed at least partially different functions to those of SEP2 in Arabidopsis, in my view, the present manuscript provides no means of choosing between Hypotheses 1 and 2 (so what was the point in even reporting these uninformative experiments?).

Validity of the findings

As I said above, the sequence and (very preliminary) expression data communicated in this paper are valid additions to what is known of plant MADS-box genes in general. We learn from these data that at least two SEP genes are present woad and that at least one of them (SEP2) is expressed in all floral organs, as is its Arabidopsis ortholog. But that is really all we learn (i.e. not very much).

Additional comments

I am very unenthusiastic about this paper. The results within it could be submitted directly to a DNA database with a brief note on the expression results- they are hardly worthy of a specific publication on their own.

Reviewer 2 ·

Basic reporting

no comment

Experimental design

no comment

Validity of the findings

The authors clarified that liSEP2 and liSEP4 are key regulators for flower development in the woad, which is a novel study. Moreover, liSEP2 and LiSEP4 could be genes related with the flower development in the woad in terms of the results of transgenic Arabidopsis. However, I could not understand what main roles of liSEP2 and LiSEP4 in the flower development from this study.

The authors described “As a cruciferous plant species, the flowers of woad are similar in structure to Arabidopsis. However, yellow petals are produced in woad flowers, and the fruits of woad are indehiscent. In addition, the fruits of woad contain only one seed and are known as silicles. Considering these different phenotypes, it makes plenty of sense to clarify the roles of woad MIKC-type MADS transcriptional factors, including SEPALLATA proteins, in controlling the flowering time and the reproductive growth.” in Abstract. These sentences were not described in the main text and not correspondent with the results of this study. The author should rewrite the Abstract to correspond with the results in this study. Additionally, the described results are not well described by the Figures. I have listed specific comments which may guide the authors through revisions.

L307-312:These sentences are unclear.

L307: Please explain why the angle of leaves and stem were changed by overexpression of liSEP2 gene in the Arabidopsis in the Discussion section.

Fig.4: I do not understand what Figure 4A-C meant in this Figure. Please explain the meanings of Figure 4A-C. It appeared that the angle of leaf in Fig. 4F and 4G looks different, both pictures are the liSEP2 overexpressed plants? What is the difference between Figure D and E? It does not make sense to me. Please compare the wild type with transgenic plant in a same picture, or same stage or same organs, or same scale of picture, if the authors highlight the differences in the flowering development in woad.

L326:The authors have described “two congenitally fused carpels that form the central gynoecium in the fourth whorl (Figs.6A-6C)”, but it’s hard to understand from Fig.6A-6C. It should be better to indicate only gynoecium like Fig. 7F.

L328: “A number of flowers carrying the wilted floral organs were male sterile (Fig. 5D)” does not make sense to me. Please rewrite and make it clearer for readers.

L329-330: Authors mentioned “At the same time, the inflorescence meristems were transformed into defective terminal flowers without sepals and petals, and the number of stamens was reduced (Figs. 5E-5F).”, but the number of stamens was not shown in Fig. 5E-F. Please show the number of stamens in Fig. 5E-F and replace the picture of flower buds with a new picture of flower stamen.

L333-334:Authors described “In addition, some apical flowers with secondary flower lacked the sepals and the petals, and the number of the stamens was decreased,”, and the authors should show the data about apical flowers with secondary flower.

L332-337: The abnormal phenotype of 35S::liSEP2 transgenic plants are shown in Fig. 6, and these phenotype are mentioned in L332-237. However, the line numbers and Figures are not corresponding. Fig. 6G is 5, 6, 7 or 8?
L332: line 7 and 8 (Fig. 6E-6G)
L334: line 3 and 4 (Fig. 6D,6F)
L337: line 5 and 6 (Fig. 6G-6H)

Fig. 6: Authors should describe “The sepals have been removed from of Arabidopsis flower.” in the legend of Fig. 6C. The explanation of Line 333-337 is not corresponding to Figure 6. The authors should add information about line number of transgenic plants in each Figure.

L363:Authors described “In some transgenic flowers, sepals are fused with petals, and stamens are fused with petals (Figs. 8C-8H)”, but it does not make sense to me from Fig. 8. It should be better to show each floral organ like Fig. 9 F, G, H, J,K and L.

L365: Please explain why the authors performed the exp. of VIGS targeting only liSEP4and ap1 cal double mutant for overexpression of liSEP4. Both liSEP2 and liSEP4 should be analyzed to understand the function of both genes on flower development in woad. The authors should mention why liSEP2 was not carried out in this experiment.

Fig S5:Why AP3 and P1 were decreased by liSEP2OX?

Additional comments

Fig. 1A, B, and C is enough as a supplementary material. Furthermore, Figure 1C is fuzzy and low quality, and unclear. Please repaint it and make it clear for understanding for readers.

Fig.9: Please add a scale bar. The picture of F, G, H, J, K and L should be better upside down.

Minor point
L131: Thalictrum thalictroides (Ranunculaeceae) -> Ranunculaeceae (Thalictrum thalictroides)
Fig S5: The asterisk where is a significant difference is missing. Please add the asterisk in the Fig S5 for making it clear.

There are some unclear points in this article, and it is necessary to show more evidence to fully support the function of liSEP genes. In my opinion, the article is interesting and presents valuable information about flower development in woad. Overall, I strongly suggest revising the text for English, and the clarity of exposition. It is also recommended to thoroughly review the manuscript and to consult with a professional translator or with someone with the relevant expertise to give guidance on English writing. Please check the flow and structures by a native English speaker to better understanding the paper. Unfortunately, English and clarity of the manuscript needed to be improved so much, and the text sounds quite unprofessional.

Reviewer 3 ·

Basic reporting

The submitted manuscript entitled that "SEPALLATA­-like genes of Isatis indigotica can affect the architecture of the inflorescences and the development of the floral organs" describes phenotypic changes of Arabidopsis and I. indigotica plants when IiSEP2 or IiSEP4 is ectopically overexpressed or silenced. Overall the manuscript is very descriptive but the phenotypic changes are well documented. English is readable enough. Because I am not a native speaker of English, I don't evaluate the language quality any more. Background information is explained adequately but some parts of Discussion should be incorporated into Introduction. Figures seem to be properly prepared except for Fig.9 which lacks scale bars.

Experimental design

Overall experimental design looks appropriate. However, the experiment about IiSEP4 in Arabidopsis may have some unclear points like no description about the number of plants used in the experiment and the number of plants which showed actual phenotype. Likewise, VIGS experiment in I. indigotica also lacks such basic information.

Validity of the findings

I thought the conclusion which the authors want to mention is properly supported by the presented data but some experimental design should be clarified as mentioned above.

Additional comments

1. L350: The authors should describe what they did and its motivation before stating the observed results.
2. L358: I couldn’t understand why the authors are motivated to transform 35S::IiSEP4-GFP onto ap1 cal double mutant. Explain the background briefly first.
3. L366: Again, the authors should mention what they were going to do and its motivation before stating the results.
4. L366: Explain what is “TRV”.
5. L374: No actual quantitative data is presented about the bolting time of the IiSEP4 silenced lines.
6. Discussion looks too long! I thought L408-419, 470-490 are not necessary.
7. I thought the statistical tests in Fig.2 and 3 need multiple testing correction.
8. No description about what kind of statistical test was employed for Fig.3.
9. Fig.9 lacks scale bars.

---

## Round 0.2 · Minor Revisions

Please address remaining issues pointed by the reviewer and amend your manuscript accordingly

Reviewer 2 ·

Basic reporting

After reading the new version of your MS, I am happy with the authors taking into account most of my previous comments, and the presentation of results cleared up significantly.

However, I am still having troubles with the scale bar of Figure 3.

I think the scale bar in all Figure 3 is misdescribed. The author indicates 1 cm for the scale bars in Figure 3A to 3C, however I don't think both image is the same size. It can be misdescribed in both 3A and 3C, or 3A or 3C. Furthermore, scale bars of 3D to 3F represent as 1mm, but they do not appear as 1 mm. If so, the plant in Arabidopsis can be too small. Please check them again, carefully and modify them for the new version of the manuscript.
Moreover, the author should replace the sentence “The sepal have been” with “The sepal and petal have been” in legend of Figure 5.

Experimental design

no comment

Validity of the findings

no comment

Additional comments

no comment

---

## Round 0.3 · Minor Revisions

Although issues were adequately addressed and the manuscript was amended accordingly, the section editor pointed out several additional points that need your attention. namely, it seems that it would benefit from some minor revisions. Some of the Latin names in the abstract are not in italics. The methods for the phylogenetic work are not providing enough detail. What phylogenetic analysis was performed in MEGA exactly? It then goes on to say that ML was performed in IQ-tree but no information is given about model parameters etc. Further, in the results, there is no discussion of any phylogenetic results. So the authors either 1) include the results of these analyses, OR 2) remove all phylogenetic reference because it doesn't seem that it was that important to the manuscript anyway.

Also, for the ANOVA and t-test methods, can the authors be more specific about what data was actually being compared/tested by each of these? Saying 'data' isn't very helpful. Finally, what post-hoc test was used for the ANOVA?.

---

## Round 0.4 · accepted · Accept

All indicated issues were adequately addressed and the manuscript was revised accordingly. Therefore, the amended version is acceptable now.

Reviewer 2 ·

Basic reporting

minor point
Columbia (italic) should be changed to Columbia (non-italic).
Ler (non-italic) should be italicized to only "er" (italic), not "Ler".

Experimental design

no comment

Validity of the findings

no comment

Additional comments

no comment